# Awareness, knowledge, and attitude towards basic life support among healthcare professional students in Bangladesh

Sohel Ahmed[1,2]*, Mohammad Jahirul Islam[3], Khandaker Md Kamrul Islam[3], Jalal Uddin[4], Farhana Khandoker[4], Tazveen Fariha[4], Progya Laboni Tina[5], Nahida Zafrin[6], Tofajjal Hossain[7], Md. Zamilur Rahman[8], Md. Zahidul Islam[9], Muhammad Hezbullah[6]

1 Ahmed Physiotherapy & Research Center, Dhaka, Bangladesh, 2 Directorate of Students' Welfare, Bangladesh University of Engineering and Technology, Dhaka, Bangladesh, 3 Department of Physical Medicine and Rehabilitation, MAG Osmani Medical College Hospital, Sylhet, Bangladesh, 4 Department of Public Health, University of Nevada, Maryland Parkway, Las Vegas, Nevada, United States of America, 5 Department of Community Medicine, Parkview Medical College, Sylhet, Bangladesh, 6 Department of Medicine, Sylhet MAG Osmani Medical college, Sylhet, Bangladesh, 7 Department of Physiotherapy and Rehabilitation, Jashore University of Science and Technology, Jashore, Bangladesh, 8 Global Dental Center, Dhaka, Bangladesh, 9 Department of Physiotherapy, Saic College of Medical Science and Technology, Dhaka, Bangladesh

* ptsohel@gmail.com

## Abstract

A comprehensive understanding of emergency care is an essential skill for all medical students to effectively manage emergency situations. The aim of this study was to evaluate healthcare professional students' understanding of basic life support (BLS) and to identify the factors that influence this knowledge. A Multi-center cross-sectional study was conducted across eight educational institutions in the Dhaka, Jashore, and Sylhet divisions of Bangladesh from January to June 2024, which included final-year medicine, physiotherapy, and nursing students. A total of 486 participants were selected by the convenience sampling method. A bi-variable and multi-variable logistic regression analysis was carried out and the p-value was set at <0.05. In this study, 73.3% of participants had inadequate knowledge in basic component (BC) and 82.5% of participant had inadequate knowledge in individual component (IC) of BLS. Medicine students demonstrated a threefold increased understanding of BLS in its BC (AOR 2.80; 95% CI 1.70–4.61, p=0.001). Participants who completed the BLS course demonstrated a twofold increase in knowledge of BLS (AOR 2.14; 95% CI 0.82–5.60, p=0.123) regarding the BC. In contrast, female students demonstrated significantly lower knowledge of the IC of basic life support (AOR 0.56; 95% CI 0.33–0.93, p=0.025). Medicine students (AOR 2.36; 95% CI 1.25–4.46, p=0.009) and nursing students (AOR 2.26; 95% CI 1.18–4.32, p=0.014) exhibited a significantly better understanding of the IC of BLS. Participants who had already taken the BLS course knew four times as much about each part of BLS (AOR

**Data availability statement:** All relevant data are within the paper and its Supporting Information files.

**Funding:** The authors received no specific funding for this work.

**Competing interests:** The authors have declared that no competing interests exist.

3.98; 95% CI 1.54–10.27, p = 0.004). This research indicates that healthcare professional students in Bangladesh possess insufficient knowledge of BLS. It is crucial for national health policymakers to priorities this finding and implement training programs for students, as well as professionals.

## Introduction

Basic life support (BLS) is a critical intervention aimed at reducing the complications related to cardiac arrest. The process entails recognizing symptoms of cardiac arrest, activating emergency response processes, and executing successful cardiopulmonary resuscitation (CPR) to restore normal blood circulation and breathing [1]. Cardiovascular diseases are the primary cause of morbidity and mortality, accounting for 57% of deaths in low- and middle-income countries [2]. A prompt intervention within 3–5 minutes after the onset of cardiac arrest is crucial for reducing mortality and morbidity [3,4]. The Global Burden of Disease Study 2019 indicated that ischemic heart disease is among the top three causes of mortality, while sudden cardiac arrest (SCA) constitutes a significant global risk factor for hospitalization [5]. Despite the advancements achieved, SCA accounts for 15%–20% of fatalities in Western societies [6]. Globally, almost 92% of individuals experiencing out-of-hospital cardiac arrest succumb owing to the lack of timely BLS interventions [7].

Medical professionals often face emergencies; hence, they must possess a comprehensive understanding of BLS [5]. Healthcare professionals, including physicians, nurses, dentists, and physiotherapists, who often face life-threatening medical emergencies, require training in BLS. Multiple factors may influence healthcare professionals' perceptions of BLS. A Japanese research indicated a decline in workers' interest in BLS training with advancing age [8]. Additional factors, including insufficient training leading to diminished confidence, apprehension over the transmission of infectious diseases via mouth-to-mouth resuscitation, and anxiety about the potential malfunction of an automated external defibrillator, contribute to a negative disposition towards doing BLS. Moreover, the experts suggested that attitude may be enhanced by consistent instruction [9].

Prior research in India indicated that medical, dentistry, and nursing students, as well as doctors and nurses, had markedly inadequate understanding of BLS [10]. Another study in Pakistan revealed that medical practitioners were deficient in skills and knowledge pertaining to BLS [7]. Adewale et al. indicated in their study that university students in Nigeria had a significant awareness and a positive disposition towards studying and performing cardiopulmonary resuscitation [11]. In contrast, a study of medical professionals in Afghanistan revealed insufficient knowledge, comprehension, and attitudes about BLS [12]. Research on younger doctors in the UK indicated their inability to execute efficient resuscitation, despite having undergone life support training [13]. These examples from various areas demonstrate a deficient understanding of BLS among healthcare professionals and undergraduates in related disciplines. However, no research has been undertaken to compare the understanding of BLS across three essential healthcare professions—doctors, physiotherapists,

and nurses—in Bangladesh, nor to assess the variables influencing their awareness. The objective of the research was to evaluate the knowledge of BLS among medical, nursing, and physiotherapy students, as well as to determine the variables influencing this knowledge among healthcare professionals. The findings would highlight the shortcomings in the existing curriculum in these areas and assist in the future development of BLS programs in Bangladesh.

## Methodology

### Ethics statement

The Institutional Ethical Committee of Physiotherapy Rehabilitation and Research, affiliated with the Bangladesh Physiotherapy Association, approved ethical approval for this study with the reference number BPA-IPRR/IRB/19/01/2023/75. The research was executed in accordance with the amended Helsinki Declaration of 2013 and the National Ethical Guidelines for Biomedical Research Involving Human Subjects, 2017. Written informed consent was obtained from the participants prior to participating in this study.

### Study design and participants

A Multi-center cross-sectional study was conducted across eight educational institutions in the Dhaka, Jashore, and Sylhet divisions of Bangladesh from January to June 2024. The research encompassed students from various health care professional institutes across three different cities, including final-year physiotherapy students from Bangladesh Health Professional Institute, Saic Institute of Medical Science and Technology, Jashore University of Science and Technology, and National Institute of Traumatology and Orthopedic Rehabilitation; final-year nursing students from Surma Nursing College and Sylhet Nursing College; and final year medicine students from Parkview Medical College and Sylhet MAG Osmani Medical College, thereby reducing selection bias. The questionnaire was administered to eight different institutions using non-probability convenience sampling.

### Inclusion and exclusion criteria

This research included final-year students in healthcare professions, including those pursuing medicine, physiotherapy, and nursing degrees. Healthcare professional students who did not consent to participate and those who submitted incomplete forms were excluded from the study.

### Sample size estimation

The sample size was calculated using the following formula: $n = z^2 P (1-P)/d^2$, where $z^2 = 1.96$ and $P = $ predicted prevalence rate of 50%, as there was no previous research in Bangladesh regarding this issue, and d represents the margin of error at 4.5%. As a result, the computation yielded a sample size of 476. The final sample comprises 500 individuals, accounting for a 5% attrition rate to reduce bias.

### Data collection procedures

This research utilized a questionnaire administered by an interviewer. To ensure consistency in the study, the lead researcher conducted a training session for all data collectors before the study commenced. A pre-testing data collecting session was conducted with five participants per data collector to identify and resolve any concerns before the start of the full-scale research, which is excluded from the main study. Eight trained data collectors with degrees in medicine or physiotherapy were recruited to gather data. Prior to this investigation, informed written consent was obtained from each participant. Participants were provided with an information sheet outlining the study procedures, as well as any expected benefits. Before the questionnaire was administered, each participant engaged in a brief interactive discussion with the data collector. The data collection process took approximately 20 minutes to complete.

### Data collection tools

An interviewer-administered questionnaire was developed for data collection purposes. The questionnaire sought to evaluate the knowledge, attitudes, and practices of BLS among final-year healthcare professional students. The questionnaire consisted of three primary sections. The initial section comprised sociodemographic information, including age and gender, and the completion of BLS and ALS training. The second section comprised eight questions pertaining to basic knowledge of BLS. The questionnaire was developed based on the American Heart Association Guidelines [14] and prior conducted studies in various nations (S1 Text). A scoring system was implemented, aggregating scores and categorizing participants into three groups: minimal knowledge (0–3), adequate knowledge (4–5), and good knowledge (6–8). The final section comprised nine questions regarding knowledge of individual components of BLS. Scores were aggregated, resulting in the classification of participants into three categories: poor knowledge (0–3 points), adequate knowledge (4–6), and good knowledge (7–9).

### Data analysis

Data analysis was conducted using IBM SPSS Statistics V.25 for Windows. For quantitative data, mean and SD were calculated, and qualitative data were presented as frequencies and percentages after being methodically collected and organized. The correlation between participants' knowledge and attitude and other variables, such as gender, discipline, BLS and ALS training courses taken, and attitude towards BLS, was evaluated using the chi-square test. The research used a multivariate logistic regression analysis to evaluate the determinants of BLS knowledge among participants. For the analysis, the basic knowledge portion was first divided into three categories and then consolidated into two: poor knowledge (0–5) and good knowledge (6–8). The individual components of BLS were also classified as poor knowledge (0–6) and good knowledge (7–9). Independent variables exhibiting p-values below 0.05 in the univariate analysis were included, and adjusted odds ratios (AORs) with 95% confidence intervals (CIs) were used for multivariate analysis. The model's efficacy in predicting BLS knowledge was assessed by the Hosmer-Lemeshow test and a categorization table. The multicollinearity among the independent variables was assessed using variance inflation factors (VIFs), with a threshold established at $VIF \leq 5.0$ [15]. A significance threshold of $p < 0.05$ was used in all cases.

## Results

### Sociodemographic characteristics of the participants

This study includes 500 final-year health professional students; among them, 486 participants were included in the final analysis, and 14 were excluded as a result of inappropriate submission. The mean age of the participants was $23.46 \pm 1.17$ years. This study includes a nearly equal number of male (50.6%) and female (49.4%) participants from physiotherapy (38.3%), medicine (32.3%), and nursing (29.4%) students. About 82.3% of the participants are familiar with the abbreviation BLS, while only 10.1% have attended BLS training and 6.6% have attended ALS training. About 75.3% of participants think everyone should have knowledge regarding BLS, and 90.1% of participants recommended it in their academic curriculum. Details are presented in Table 1.

### Basic component knowledge of BLS among the participants

Among the study participants, 73.3% had poor knowledge and 26.7% reported good knowledge in basic components. One-fourth (26.5%) of the participants said that BLS can only be performed in a hospital setting. Approximately 43.0% of participants reported witnessing the procedure of BLS, while 55.6% acknowledged a lack of understanding of the management of an unconscious person. The emergency dialing number was incorrectly answered by 13.0% of participants, and 6.6% of were unaware of it. Nearly half of the participants (49.2%) don't know when to start CPR, and 38.5% of the

PLOS Global Public Health

**Table 1. Socio-demographic characteristics and attitude towards BLS among participants (n = 486).**

| Variables | Frequency | Percentage |
|---|---|---|
| **Gender** | | |
| Male | 246 | 50.6 |
| Female | 240 | 49.4 |
| Discipline | | |
| Physiotherapy | 186 | 38.3 |
| Medicine | 143 | 32.3 |
| Nursing | 157 | 29.4 |
| **Attained BLS training** | | |
| Yes | 49 | 10.1 |
| No | 437 | 89.9 |
| **Attained ALS training** | | |
| Yes | 32 | 6.6 |
| No | 454 | 93.4 |
| **Know the abbreviation of BLS** | | |
| Know | 400 | 82.3 |
| Don't know | 86 | 17.7 |
| **Do you think everyone should have the knowledge regarding BLS?** | | |
| Yes | 366 | 75.3 |
| No | 120 | 24.7 |
| **Do you recommend BLS in your academic curriculum?** | | |
| Yes | 438 | 90.1 |
| No | 48 | 9.9 |

participants provided incorrect answers regarding the question: CPR should start within... seconds. Nearly half of the participants (48.5%) don't know how to check the pulse of an unconscious patient. Refer to Table 2 for details.

### Individual component knowledge of BLS

Among the study participants, 82.5% had poor knowledge and 17.5% reported good knowledge in individual components. Over fifty percent of the participants are aware with the cardiac compression rate, but 59.7% are unfamiliar with the ratio of cardiac compressions to breath supply rate. Fifty percent of the participants lack knowledge on the rescue breath rate, and 56.0% are uncertain about the appropriate location for doing chest compressions. The appropriate depth for adult chest compressions was answered by 64.0% of the participants. Only 20% of the participants executed the BLS autonomously. The extent of pulse checks was unknown to 37.4% of participants, while the frequency of pulse re-checks was unknown to 59.5% of the participants. Information is provided in Table 3.

Among the study participants, 34.2% reported the reasons for the lack of BLS knowledge were due to non-availability of professional training, 14.2% due to lack of interest, 12.3% due to busy curriculum, and 39.1% due to all of the factors (Fig 1). 41.4% of participants graded their BLS knowledge as poor, 38.9% as below average, 16.5% as good, and only 3.3% as excellent (Fig 2).

### Association between BLS knowledge and socio-demographics of participants

Basic understanding of BLS correlates with gender (p = 0.001), discipline (p < 0.001), and prior completion of BLS training (p = 0.008). We identified a strong correlation between the variable of everyone should have knowledge regarding BLS

**Table 2. Knowledge of basic components of basic life support among the participants (n = 486).**

| Variables | Frequency | Percentage |
|---|---|---|
| **BLS can be performed at** | | |
| Only hospital setup | 129 | 26.5 |
| Both inside and outside the hospital | 357 | 73.5 |
| **Observed BLS being performed** | | |
| Yes | 209 | 43.0 |
| No | 277 | 57.0 |
| **If you don't want to give mouth-to-mouth CPR, what can be done?** | | |
| Incorrect answer | 270 | 55.6 |
| Correct answer | 216 | 44.4 |
| **What do you do first when you get an unconscious person?** | | |
| Know | 216 | 44.4 |
| Don't know | 270 | 55.6 |
| **Know the emergency dialing number** | | |
| Don't know | 31 | 6.4 |
| Correct | 392 | 80.7 |
| Incorrect | 63 | 13.0 |
| **How do you know when to start compression?** | | |
| Know | 239 | 50.8 |
| Don't know | 247 | 49.2 |
| **CPR should start within ……. seconds of recognition of cardiac arrest.** | | |
| Incorrect answer | 187 | 38.5 |
| Correct answer | 299 | 61.5 |
| **How do you check the pulse of an unconscious person?** | | |
| Know | 251 | 51.6 |
| Don't know | 235 | 48.4 |

(p < 0.001) and the suggestion for the incorporation of BLS into the academic curriculum (p < 0.001). Significant correlations existed between the participants' self-evaluations of their BLS knowledge (p = 0.002) and their self-ratings of that knowledge (p < 0.001). The participants' previous completion of ALS training showed no significant correlation. Refer Table 4 for details.

## Results from regression analysis

Female students exhibited superior understanding in the basic component of BLS (AOR 1.12; 95% CI 0.72–1.74, p = 0.624); however, they were lacking in understanding in individual component of BLS (AOR 0.56; 95% CI 0.33–0.93, p = 0.025). Medicine students possessed approximately threefold greater understanding of the fundamental elements of BLS (AOR 2.80; 95% CI 1.70–4.61, p = 0.001). Individuals who finished the BLS course demonstrated a twofold increase in knowledge of BLS (AOR 2.14; 95% CI 0.82–5.60, p = 0.123). Participants who believed that everyone should have BLS knowledge (AOR 1.38; 95% CI 0.77–2.49, p = 0.285), as well as those who supported the inclusion of BLS training in academic curricula (AOR 2.61; 95% CI 0.85–8.05, p = 0.097), exhibited greater knowledge of BLS.

Medicine students (AOR 2.36; 95% CI 1.25–4.46, p = 0.009) and nursing students (AOR 2.26; 95% CI 1.18–4.32, p = 0.014) demonstrated a significantly greater understanding of the individual component of BLS. Individuals who previously participated in the BLS course demonstrated a fourfold increase in individual component knowledge of BLS (AOR 3.98; 95% CI 1.54–10.27, p = 0.004). Those who believed that everyone should have BLS knowledge (AOR 1.78; 95%

**Table 3. Knowledge of individual components of BLS among the participants (n = 486).**

| Variables | Frequency | Percentage |
|---|---|---|
| **What is the compression rate of external cardiac massage per minutes during BLS?** | | |
| Incorrect answer | 212 | 43.6 |
| Correct answer | 274 | 56.4 |
| **What is the ratio of cardiac compression to breath delivery during BLS?** | | |
| Incorrect answer | 290 | 59.7 |
| Correct answer | 196 | 40.3 |
| **What is the rescue breathing rate per minute?** | | |
| Incorrect answer | 228 | 46.9 |
| Correct answer | 258 | 53.1 |
| **Location of chest compression** | | |
| Incorrect answer | 272 | 56.0 |
| Correct answer | 214 | 44.0 |
| **Sequence to be followed while performing BLS** | | |
| Incorrect answer | 336 | 69.1 |
| Correct answer | 150 | 30.9 |
| **Depth of chest compression in adults during CPR** | | |
| Incorrect answer | 175 | 36.0 |
| Correct answer | 311 | 64.0 |
| **Performed BLS by self** | | |
| Yes | 97 | 20.0 |
| No | 389 | 80.0 |
| **For how long the pulse should be checked?** | | |
| Incorrect answer | 182 | 37.4 |
| Correct answer | 304 | 62.6 |
| **Pulse should be re-checked on every** | | |
| Incorrect answer | 289 | 59.5 |
| Correct answer | 197 | 40.5 |

CI 0.83–3.82, p = 0.139) and those who recommended BLS in academic curriculum (AOR 2.80; 95% CI 0.61–12.99, p = 0.190) exhibited superior knowledge of individual BLS components. Details are presented in Table 5.

## Discussion

Medical students and graduates must have a fundamental understanding of emergency care to appropriately address medical emergencies. This research represents the first cross-sectional investigation into the knowledge, attitudes, and practices of BLS among healthcare professional students, specifically those in the fields of medicine, physiotherapy, and nursing, within the context of Bangladesh. This study demonstrates that healthcare students in Bangladesh exhibit a lack of adequate understanding necessary for the execution of BLS. It is imperative that national health policymakers give precedence to these findings and establish training programs for healthcare professionals, students, and the broader adult population.

The present research indicated that 73.3% exhibited inadequate knowledge of basic components, while 82.5% showed poor knowledge of individual components; just 26.7% and 17.5% had adequate knowledge of the basic and individual components of basic life support, respectively. Research performed in Ethiopia involving graduating health science and

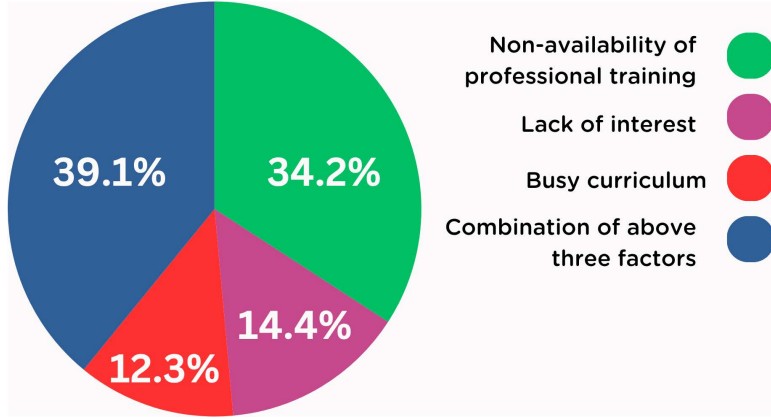

**Fig 1. Self-assessment of reasons for the lack of BLS knowledge.**

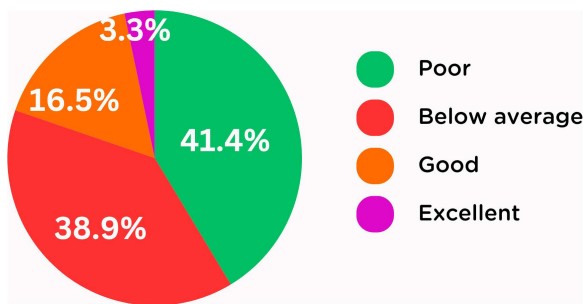

**Fig 2. Self-grading of BLS knowledge.**

medical students indicated that 56.9% possess proficient understanding and 51.5% exhibit good practice concerning basic life support, surpassing the findings of our present investigation [16]. Similar to our findings, Nepali healthcare workers possess inadequate knowledge (only 12% had adequate knowledge) regarding basic life support, which was conducted in a working hospital in Nepal [17].

In the current investigation, a small proportion of medical students (10.1%) had engaged in a training course related to BLS, whereas 75.3 percent expressed support for the integration of a specialized course into the academic curriculum. In Oman, an investigation involving medical students indicated that 35.5% of the participants had previously undergone BLS training [18], a statistic that exceeds the findings of the present study. Previous research conducted in Afghanistan indicated that 50.9% of medical graduates had received BLS training prior to their graduation [12], while 31% of Nepali medical professionals obtained BLS training subsequent to their graduation [19]. However, in a separate study carried out in Germany, the majority of participants indicated that they had engaged in BLS training [20]. Basic life support knowledge is deficient among Bangladeshi final-year healthcare professional students, necessitating essential steps to incorporate it into their academic curriculum.

This research reveals that graduate medicine students possess a superior comprehension of both the fundamental and individual elements of basic life support, whereas graduate nursing students have a heightened awareness of the individual components compared to physiotherapy students. The result aligned with the research done in Egypt, which indicated a statistically significant disparity in basic life support knowledge ratings across several disciplines [1]. Enhanced

**Table 4. Association between basic life support knowledge and the sociodemographic of the participants (n = 486).**

| Variables | Basic knowledge of BLS | | | P value | Knowledge of individual components of BLS | | | P value |
|---|---|---|---|---|---|---|---|---|
| | Minimal knowledge | Adequate knowledge | Good knowledge | | Minimal knowledge | Adequate knowledge | Good knowledge | |
| Gender | | | | | | | | |
| Male | 49 (10.1) | 125 (25.7) | 72 (14.8) | 0.001 | 96 (19.8) | 96 (19.8) | 54 (11.1) | 0.027 |
| Female | 86 (17.7) | 96 (19.8) | 58 (11.9) | | 111 (22.8) | 98 (20.2) | 31 (6.4) | |
| Discipline | | | | | | | | |
| Physiotherapy | 39 (8.0) | 107 (22.0) | 40 (8.2) | <0.001* | 75 (15.4) | 92 (18.9) | 19 (3.9) | <0.001 |
| Nursing | 79 (16.3) | 55 (11.3) | 23 (4.7) | | 88 (18.1) | 40 (8.2) | 29 (6.0) | |
| Medicine | 17 (3.5) | 59 (12.1) | 67 (13.8) | | 44 (9.1) | 62 (12.8) | 37 (7.6) | |
| Attended BLS training before | | | | | | | | |
| Yes | 5 (1.0) | 27 (5.6) | 17 (3.5) | 0.008* | 6 (1.2) | 25 (5.1) | 18 (3.7) | <0.001 |
| No | 130 (26.7) | 194 (39.9) | 113 (23.3) | | 201 (41.4) | 169 (34.8) | 67 (13.8) | |
| Attended ALS training before | | | | | | | | |
| Yes | 5 (1.0) | 20 (4.1) | 7 (1.4) | 0.135* | 4 (0.8) | 19 (3.9) | 9 (6.6) | 0.002* |
| No | 130 (26.7) | 201 (41.4) | 123 (25.3) | | 203 (41.8) | 175 (36.0) | 76 (15.6) | |
| Everyone should have the knowledge regarding BLS | | | | | | | | |
| Yes | 84 (17.3) | 173 (35.6) | 109 (22.4) | <0.001* | 138 (28.4) | 153 (31.5) | 75 (15.4) | <0.001* |
| No | 51 (10.5) | 48 (9.9) | 21 (4.3) | | 69 (14.2) | 41 (8.4) | 10 (2.1) | |
| Recommended BLS in academic curriculum | | | | | | | | |
| Yes | 109 (22.4) | 203 (41.8) | 126 (25.9) | <0.001* | 178 (36.6) | 177 (36.4) | 83 (17.1) | 0.008* |
| No | 26 (5.3) | 18 (3.7) | 4 (0.8) | | 29 (6.0) | 17 (3.5) | 2 (0.4) | |
| Self-assessment of reasons for the lack of BLS knowledge | | | | | | | | |
| Non-availability of professional training | 56 (11.5) | 81 (16.7) | 29 (6.0) | 0.002* | 65 (13.4) | 71 (14.6) | 30 (6.2) | 0.125* |
| Lack of interest | 21 (4.3) | 29 (6.0) | 20 (4.1) | | 32 (6.6) | 33 (6.8) | 5 (1.0) | |
| Busy Curriculum | 22 (4.5) | 23 (4.7) | 15 (3.1) | | 30 (6.2) | 21 (4.3) | 9 (1.9) | |
| Combination of above all three factors | 36 (7.4) | 88 (18.1) | 66 (13.6) | | 80 (16.5) | 69 (14.2) | 41 (8.4) | |
| Self-grading of BLS knowledge level | | | | | | | | |
| Poor | 85 (17.5) | 89 (18.3) | 27 (5.6) | <0.001* | 119 (24.5) | 67 (13.8) | 15 (3.1) | <0.001* |
| Below average | 38 (7.8) | 82 (16.9) | 69 (14.2) | | 77 (15.8) | 82 (16.9) | 30 (6.2) | |
| Good | 9 (1.9) | 38 (7.8) | 33 (6.8) | | 8 (1.6) | 36 (7.4) | 36 (7.4) | |
| Excellent | 3 (0.6) | 12 (2.5) | 1 (0.2) | | 3 (0.6) | 9 (1.9) | 4 (0.8) | |

awareness of Basic Life Support (BLS) among students in disciplines that include emergency patient management, emergency care, and related courses in their curriculum, along with increased exposure to emergency practices, may account for the discrepancies in knowledge among students from various disciplines. A significant number of the participants in this study, approximately 75.3%, exhibit a positive attitude towards basic life support. which is similar to other studies done in Syria, Iraq, Saudi Arabia, and Jordan [1,21].

## Clinical significance for public health

This cross-sectional research emphasizes the need for implementing compulsory basic life support training for both healthcare professional students and all graduate-level students. These results provide policymakers a foundation for establishing a fundamental life support education and training system that will cultivate a proficient workforce, potentially enhancing patient outcomes and decreasing the incidence of unexpected death within the community. Public health

**Table 5. Binary logistic regression model: predictors of basic life support knowledge of basic and individual components (n = 486).**

| Variables | AOR | 95% CI | | P value |
|---|---|---|---|---|
| | | Lower | Upper | |
| **Basic component** | | | | |
| Gender | | | | |
| Male | Reference | | | |
| Female | 1.12 | 0.72 | 1.75 | 0.624 |
| Discipline | | | | |
| Physiotherapy | Reference | | | |
| Nursing | 0.60 | 0.34 | 1.06 | 0.75 |
| Medicine | 2.80 | 1.70 | 4.61 | **0.001*** |
| Attended BLS training | | | | |
| No | Reference | | | |
| Yes | 2.14 | 0.82 | 5.60 | 0.123 |
| Everyone should have BLS knowledge | | | | |
| No | Reference | | | |
| Yes | 1.38 | 0.77 | 2.49 | 0.285 |
| Recommended BLS in academic curriculum | | | | |
| No | Reference | | | |
| Yes | 2.61 | 0.85 | 8.05 | 0.097 |
| **Individual component** | | | | |
| Gender | | | | |
| Male | Reference | | | |
| Female | 0.56 | 0.33 | 0.93 | **0.025*** |
| Discipline | | | | |
| Physiotherapy | Reference | | | |
| Nursing | 2.26 | 1.18 | 4.32 | **0.014*** |
| Medicine | 2.36 | 1.25 | 4.46 | **0.009*** |
| Attended BLS training | | | | |
| No | Reference | | | |
| Yes | 3.98 | 1.54 | 10.27 | **0.004*** |
| Everyone should have BLS knowledge | | | | |
| No | Reference | | | |
| Yes | 1.78 | 0.83 | 3.82 | 0.139 |
| Recommended BLS in academic curriculum | | | | |
| No | Reference | | | |
| Yes | 2.80 | 0.61 | 12.99 | 0.190 |

initiatives must concentrate on these domains to alleviate the effects of sudden cardiac arrest and reduce mortality risk, ultimately improving the overall health of the population.

## Strengths, limitations, and future recommendations

To the best of our knowledge, this is the first study to investigate the knowledge and practice of basic life support among graduate-level healthcare professional students in Bangladesh. This research included an interviewer-administered questionnaire, enhancing data accuracy. Participants from diverse fields and institutions enhanced the generalizability of the

results. This research used Google Forms, an ecologically friendly application that facilitates the collection of precise data without missing items from participants. Consequently, this study has certain limitations. This research used a cross-sectional design and employed a nonrandom convenience sampling method to choose participants. Most participants correctly answered the question about the depth of chest compression, possibly due to the perplexing option provided. We found no correlation between BLS knowledge and ALS training because of their significantly skewed values. Since this study used a cross-sectional method, we are unable to assess the participants' skills. We scored one point for each correct answer, but some questions were more important than others, which was another limitation. Future research should examine the effects of basic life support training on the knowledge and practices of healthcare professional students and practitioners.

## Conclusion

This survey indicates that two-thirds of healthcare professional students had insufficient knowledge and skills in BLS. The majority of participants had not undergone BLS training. Most of the participants have positive attitudes toward BLS. Training in basic life support was substantially correlated with a good knowledge score. Health institutions advocate for uniform training and evaluations to improve BLS knowledge. Basic Life Support must be regarded as an essential component for all healthcare professionals and should be integrated into the curricula of medical and health science faculties. National health policymakers need to prioritize these findings by organizing training programs for healthcare professionals and students.

## Supporting information

**S1 Checklist. STROBE Checklist.**
(DOCX)

**S1 Text. Questionnaire.**
(DOCX)

**S1 Data. Basic life support data.**
(SAV)

## Acknowledgments

The authors are grateful to all the study participants who spent their valuable time.

## Author contributions

**Conceptualization:** Sohel Ahmed, Mohammad Jahirul Islam, Khandaker Md Kamrul Islam, Jalal Uddin, Farhana Khandoker, Tazveen Fariha, Progya Laboni Tina, Nahida Zafrin, Md. Zamilur Rahman, Md. Zahidul Islam, Muhammad Hezbullah.

**Data curation:** Mohammad Jahirul Islam, Khandaker Md Kamrul Islam, Jalal Uddin, Farhana Khandoker, Progya Laboni Tina, Nahida Zafrin, Tofajjal Hossain, Md. Zamilur Rahman, Md. Zahidul Islam.

**Formal analysis:** Sohel Ahmed.

**Methodology:** Sohel Ahmed, Mohammad Jahirul Islam, Khandaker Md Kamrul Islam, Jalal Uddin, Farhana Khandoker, Tazveen Fariha, Progya Laboni Tina, Nahida Zafrin, Tofajjal Hossain, Muhammad Hezbullah.

**Software:** Sohel Ahmed, Jalal Uddin, Farhana Khandoker.

**Supervision:** Muhammad Hezbullah.

**Validation:** Sohel Ahmed, Mohammad Jahirul Islam, Khandaker Md Kamrul Islam, Jalal Uddin, Farhana Khandoker, Tazveen Fariha, Progya Laboni Tina, Nahida Zafrin, Tofajjal Hossain, Md. Zamilur Rahman, Md. Zahidul Islam, Muhammad Hezbullah.

**Visualization:** Mohammad Jahirul Islam, Khandaker Md Kamrul Islam, Jalal Uddin, Farhana Khandoker, Tazveen Fariha, Progya Laboni Tina, Nahida Zafrin, Tofajjal Hossain, Md. Zamilur Rahman, Md. Zahidul Islam, Muhammad Hezbullah.

**Writing – original draft:** Sohel Ahmed, Mohammad Jahirul Islam, Khandaker Md Kamrul Islam, Jalal Uddin, Farhana Khandoker, Tazveen Fariha, Progya Laboni Tina, Nahida Zafrin, Tofajjal Hossain, Md. Zamilur Rahman, Md. Zahidul Islam, Muhammad Hezbullah.

**Writing – review & editing:** Sohel Ahmed.

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
