## [Decision Letter · Decision Letter 0]

9 Jun 2025

PGPH-D-25-00200

Awareness, knowledge and attitude towards basic life support among healthcare professional students in Bangladesh

Dear Dr. Ahmed,

Thank you for submitting your manuscript to PLOS Global Public Health. After careful consideration, we feel that it has merit but does not fully meet PLOS Global Public Health’s publication criteria as it currently stands. Therefore, we invite you to submit a revised version of the manuscript that addresses the points raised during the review process.

When revising the manuscript please pay close attention to the reviewers' comments on both methodology and analysis and on the structure of the article, flow and presentation. Please make sure to clarify the technical aspects of the study (for example the significance of gender, height, weight and BMI in this study), the study design, data collection procedures and tools. When revising the results and discussion, please make sure to address reviewers' comments and limitations and future studies. 

Please submit your revised manuscript by . If you will need more time than this to complete your revisions, please reply to this message or contact the journal office at globalpubhealth@plos.org. Please include the following items when submitting your revised manuscript:

We look forward to receiving your revised manuscript.

Kind regards,

Anat Rosenthal

Academic Editor

Journal Requirements:

1. Please insert an Ethics Statement at the beginning of your Methods section, under a subheading 'Ethics Statement'.

2. Please provide separate figure files in .tif or .eps format.

3. We note that your Data Availability Statement is currently as follows: “Data generated and analysed during this study is included in the article and its supplementary information files.”.

Additional Editor Comments (if provided):

Reviewers' comments:

Reviewer's Responses to Questions

**Comments to the Author**

1. Does this manuscript meet PLOS Global Public Health’s publication criteria?

Reviewer #1: Partly

Reviewer #2: Partly

2. Has the statistical analysis been performed appropriately and rigorously?

Reviewer #1: Yes

Reviewer #2: Yes

3. Have the authors made all data underlying the findings in their manuscript fully available (please refer to the Data Availability Statement at the start of the manuscript PDF file)?

Reviewer #1: Yes

Reviewer #2: Yes

4. Is the manuscript presented in an intelligible fashion and written in standard English?

Reviewer #1: No

Reviewer #2: Yes

Reviewer #1: General comment -Need to work on English language.

Some suggestions in language:

Abstract - results - "The current study revealed...lacking knowledge". This sentence need grammar correction.

Introduction - Line 7-12 appears to be out of context and does not gel with the whole paragraph. This needs to be re written.

Data collection procedures - Line 3 - "A feminisation data collection session..". I don't understand what feminization means here.

Data analysis - Line 4 - "...such as gender, faculty..". What is meant by faculty?

In multiple sentences in Results section, the sentence starts with a number. This needs to be corrected. It is better not to start a sentence with a number.

Results from regression analysis- Line 5 - need to re organise the sentence to highlight the significance of this.

Individual component knowledge of BLS - Line 6 - " Merely 20% ...". This sentence needs restructuring

Discussion - line 27 - "Bangladeshi final year ...'lacings' basic life support..."

Discussion - line 38 - "In the study..have positive ". This line needs to be modified.

Conclusion- Last line needs to be modified.

References 1 and 22 are same, 12 and 19 are same

Technical aspects:

A. Questionnaire

1. What is the significance of gender, height, weight and BMI in this study?

2. Kindly provide the answers of question 18 and 19 with references

3. Depth of CC - The total response is less than 486. Also, the result is hard to accept given the fact that most of the answers on basic questions like CPR rate and frequency were answered incorrectly while depth of CPR mostly had correct responses. The doubt further intensifies looking at the confusing options provided for the question

B. BLS knowledge and ALS training correlation is not found because of the highly skewed value. This need to be acknowledged.

C. Limitations - This is a cross sectional survey. No skill was assessed in the study. This limitation needs to be acknowledged. Also there is a possibility of bias inherent to all studies based on survey. Also, I was unable to understand the method of survey. Was it individually completed or as a group? Were they allowed to discuss or refer to any materials while completing the questionnaire?

D. How was the scores of different components decided? One component/ question of the questionnaire might carry higher weightage compared to another one. Was this considered during scoring?

Reviewer #2: The authors address an important subject (the knowledge of Basic Cardiac Life Support [BCLS] among final year students in 3 relevant health professions) in Bangladesh and the authors are to be complimented for addressing it.

Abstract

It would be important to note that three professions (medicine physio and nursing) are being examined. This is clear later in the text, but that is not early enough.

Introduction

A brief description of the three programs is in order. Are these post-graduate or undergraduate courses, how many years of study, etc ? A small table comparing the three professions would be of use to the reader.

I suggest you delete the last two lines of the introduction “We anticipate… for healthcare professionals." Such sentiments belong in the discussion or conclusion.

Study design

How were the eight study institutions chosen ? How many such in Bangladesh ? Do they represent (a more or less) representative sample of institutions teaching the three professions? Public versus private ? size ? etc. If websites are available for the institutions involved, they should be added

“MBBS“ should be changed to "medicine" throughout.

Data collection procedures

Please explain “feminisation data collection session“

How many data collectors were there?

Data collection tools

It would be interesting for the reader to be able to read the questionnaires. These could be included in an appendix. With respect to sociodemographic information, it is unclear why weight, height and BMI are included as they seem irrelevant to the study question.

Results

One certainly does not need two decimal points for most of the data presented. This implies a level or resolution irrelevant to the data.

The tables take up a lot of space and you have adequately summarised much of the relevant data in the text. They could be shortened or deleted

You should add the total number of participants in the legends. Eg in

table 1 “… towards BLS among 486 participants“

“figure one and two" are mentioned in the text but I cannot find them

p 13 Results from regression analysis:

Information in the text with respect to female students does not seem to be consistent with what is seen in the table; please clarify.

Discussion

I suggest the first sentence reads: “A fundamental understanding of emergency care is an essential competency for all health professionals, especially for those practicing the three disciplines examined here. “

Page16 - in the middle of the text the word “nevertheless" seems extraneous.

Strengths, limitations and future recommendations:

I suggest that the first sentence read “To the best of our knowledge, this research is the first…“. One never knows if there is something else in the literature that examined what you did.

Tables and text: I would change the word “department“ to “discipline“ throughout

**Do you want your identity to be public for this peer review?** For information about this choice, including consent withdrawal, please see our Privacy Policy

Reviewer #1: No

Reviewer #2: No

---

## [Decision Letter · Decision Letter 1]

1 Aug 2025

PGPH-D-25-00200R1

Awareness, knowledge and attitude towards basic life support among healthcare professional students in Bangladesh

Dear Dr. Ahmed,

Thank you for submitting your manuscript to PLOS Global Public Health. After careful consideration, we feel that it has merit but does not fully meet PLOS Global Public Health’s publication criteria as it currently stands. Therefore, we invite you to submit a revised version of the manuscript that addresses the points raised during the review process.

Following the reviewers’ feedback, we kindly request that you review and revise the manuscript for grammatical mistakes to improve language and editorial quality.

We look forward to receiving your revised manuscript.

Kind regards,

Anat Rosenthal

Academic Editor

Journal Requirements:

Reviewers' comments:

Reviewer's Responses to Questions

**Comments to the Author**

Reviewer #1: All comments have been addressed

publication criteria?

Reviewer #1: Partly

3. Has the statistical analysis been performed appropriately and rigorously?

Reviewer #1: Yes

4. Have the authors made all data underlying the findings in their manuscript fully available (please refer to the Data Availability Statement at the start of the manuscript PDF file)?

Reviewer #1: Yes

5. Is the manuscript presented in an intelligible fashion and written in standard English?

Reviewer #1: No

Reviewer #1: There are still some grammatical mistakes which needs to be addressed.

**Do you want your identity to be public for this peer review?** For information about this choice, including consent withdrawal, please see our Privacy Policy

Reviewer #1: **Yes: ** Vineeth Chandran K P

---

## [Editor Report · Decision Letter 2]

8 Aug 2025

Awareness, knowledge, and attitude towards basic life support among healthcare professional students in Bangladesh

PGPH-D-25-00200R2

Dear Dr. Ahmed,

We are pleased to inform you that your manuscript 'Awareness, knowledge, and attitude towards basic life support among healthcare professional students in Bangladesh' has been provisionally accepted for publication in PLOS Global Public Health.

Best regards,

Anat Rosenthal

Academic Editor